# Digital Transformation and Export Quality of Chinese Products: An Analysis Based on Innovation Efficiency and Total Factor Productivity

**Fei Wang [1,2] and Linwei Ye [1,*]**

1   School of Economics and Trade, Guangdong University of Foreign Studies, Guangzhou 510006, China
2   School of Economics and Trade, Guangzhou Xinhua University, Guangzhou 510006, China
*   Correspondence: 20220220008@gdufs.edu.cn

**Abstract:** In recent years, Chinese manufacturing enterprises have competed to chase the wave of the "digital revolution"; digital empowerment has become an important strategic path of technological reform for many manufacturing enterprises. Based on the micro data of listed companies, this paper investigates the impact of digital transformation of Chinese listed companies on the quality of their export products. It is found that digital transformation can significantly improve the quality of enterprises' export products. After a series of robustness tests, this conclusion is still valid, and there are two ways to improve innovation performance and total factor productivity. The export product quality enhancement effect of enterprise digital transformation also has the heterogeneity of ownership, region, and industry. Furthermore, this paper also examines the impact of digital transformation on the internal salary gap of enterprises, and finds that digital transformation may increase the internal salary gap of enterprises and form a "masking effect" on the quality of export products. To a certain extent, this paper deepens the understanding of the study of enterprise digital transformation on the quality of export products and its differences and provides certain guidance for enterprises to implement the strategy of digital transformation.

**Keywords:** digital transformation; innovation performance; total factor productivity; quality improvement

## 1. Introduction

Since the country's accession to the WTO, China's foreign trade has achieved leapfrog development. Through processing trade, China has participated in the global industrial division of labor and promoted the initial development of the manufacturing industry. However, in recent years, China's labor costs have risen, traditional labor comparative advantages have weakened, the internal and external environment is complicated and severe, external demand is depressed, and trade frictions are frequent. As a result, the long-term explosive growth rate of the foreign trade volume has slowed down, and the competition among products in the international product market has shifted from price competition to quality competition. The quality upgrade of export products is a key link for Chinese manufacturing industries to maintain their competitiveness in international markets. Digital economy continuously injects new impetuses into traditional economy and has become an important driving force for national economic development [1]. The rapid development of big data, artificial intelligence, cloud computing, and other information technologies has driven a new round of industrial replacement and triggered profound changes in international trade [2]. In 2021, the value added of digital economy in 47 major countries reached US $38.1 trillion, a nominal increase of 15.6 percent year on year, accounting for 45.0 percent of GDP. The rapid development of digital economy improves the production efficiency of enterprises, shortens the life cycle of products, and deepens the transformation of product structure. This is certainly an opportunity for trading companies in developing countries seeking to climb the global value chain [3]. Especially

for developing countries with a huge market such as China, digital economy is not only conducive to the innovative development of foreign trade, but also promotes the formation of a "double cycle" development pattern [4], promotes China's Internet to enter a new stage from a consumer Internet to an industrial Internet. It is urgent to improve the high-quality development of foreign trade. Digital empowerment is conducive to promoting the digital transformation of China's manufacturing enterprises, achieving high-quality development of trade through export product upgrading, and improving the country's competitiveness in the global industrial value chain [5]. To study how foreign trade manufacturing enterprises, as micro subjects promoting digital transformation under the new situation of accelerating the promotion of the digital enabling manufacturing industry, can effectively release the effect of digital transformation on the quality improvement of export products and how the effect mechanism of digital transformation impacts the quality improvement of export products from China's experience can provide a strong reference value for developing countries in improving their trade competitiveness.

Based on the existing literature, factors influencing the quality of export products of enterprises and the quality effect of export products of enterprises' digital transformation are investigated. The relevant research closely related to the issues concerned in this paper is mainly as follows.

First of all, there is related research on the quality of export products. As the key to the transformation of foreign trade growth pattern and the optimization of export structure, export product quality concerns whether China can realize high-quality development of trade. A new trade theory indicates that export enterprises have high production efficiency and low production cost, so it is proposed that only enterprises with high production efficiency will export [6], while the latest trade theory integrates the heterogeneity of product quality on the basis of a traditional model [7]. The relationship between product export quality, transportation cost, price, and productivity is analyzed using product and enterprise level data. Enterprises with high productivity will choose to use high-quality inputs to produce high-quality products for export to developed countries [8]. Secondly, because the quality level of export products is a latent variable that is difficult to estimate, the quality level of export products is estimated mainly through product technical complexity [9] and product use value [10]. The utility function of consumers is introduced into the model derivation process, and the average utility is used to measure the product quality level [11]. Finally, in the relevant research on how to improve the quality of export products, it is found that the production of higher quality varieties of existing products by export enterprises can help build the existing comparative advantages to improve export income and productivity, while the quality of institutions, free trade policies, foreign direct investment inflows [12], and human capital can promote the quality upgrade of export products [13].

Secondly, there is research on the definition and measurement of digital transformation. The category of digital economy, including digital hardware and software infrastructure [14], digital business network and organization, and products traded in e-commerce, lays the foundation for the measurement of enterprises' digital transformation [15]. Relevant research on digitalization, digitization, digitalization and digital transformation includes three different development stages [16] (see Figure 1). Digital transformation emphasizes enhancing the core competitiveness of enterprises in the business environment market by developing new businesses [17]. Digital transformation will make use of digital resources to generate differentiated value [18], promote the exchange and communication between enterprises, so as to promote manufacturing enterprises to improve and create production processes and products, and enhance their technological innovation performance [19]. Digital technology, innovation, and skills are interdependent. Businesses need new skills to innovate, learn, and adapt to evolving digital technologies, which in turn require changes in the codification of knowledge in order to carry out productive and innovative activities. In addition to using the density of industrial robots to measure the indicators of different digital technologies [20], the influence and mechanism of digital

empowerment on the quality of export products are discussed in depth. In addition, the annual reports of listed companies in different years can be used to measure the digital transformation of enterprises by word frequency statistics [21].

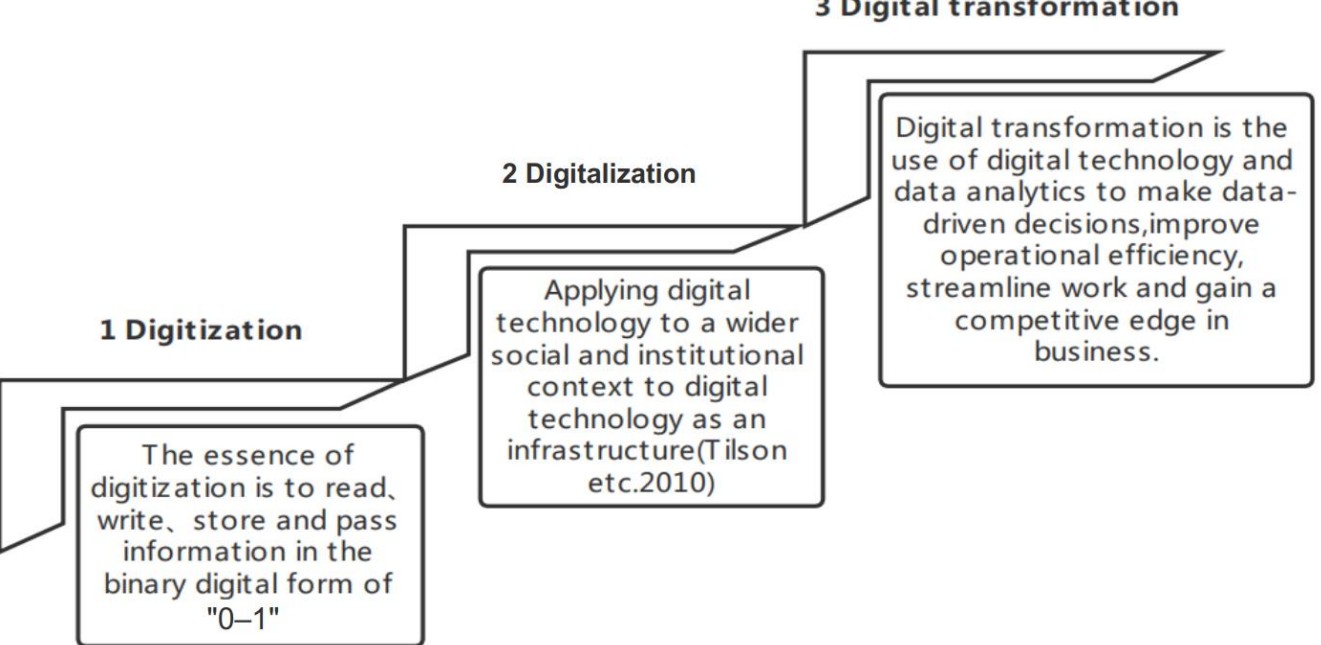

**Figure 1.** Digital transformation and development stage.

Finally, there is research on the influence of enterprise digital transformation on foreign trade. Digital economy is an important driving force to promote the high-quality development of the manufacturing industry, which can improve the export competitiveness of the manufacturing industry [22]. The intensity of digital transformation can significantly improve the quality of export products [23]; digital transformation mainly affects the domestic value-added rate of manufacturing exports [1] and further improves the product quality of enterprises, thus enhancing the international competitiveness [24]. On the other hand, the development of digital technology can reduce the costs of foreign trade enterprises in five aspects, namely, search cost, reproduction cost, transportation cost, tracking cost, and verification cost [25]. Among them, the reduction of search cost can improve the efficiency of information matching and information communication and organization [26], reducing the information cost and search cost of enterprises in international trade [23], so as to improve the quality of enterprises' export products [20].

Combing through the existing literature, it is found that there are many studies on the measurement of export product quality and enterprise digital transformation, and most of the studies argue that digital technology can significantly improve enterprises' trade mode and enhance the quality of export products. However, on the whole, there are the following deficiencies: First of all, Khandelwal's [20] approach is generally adopted to measure the quality of export products, but the results vary greatly and may be negative, which cannot truly reflect the quality level of enterprise products. Secondly, in the measurement of enterprise digital transformation, management decision-making is an important factor affecting enterprise digital transformation, but the current digital transformation indicators rarely consider management decision-making factors. Finally, the existing literature has carried out a theoretical review of the export product quality effect of enterprise digital transformation, but there is a lack of empirical discussion on the mechanism verification and analysis. The marginal contribution of this paper lies in the following aspects: In terms of method measurement, this paper calculates the export product quality results at the enterprise-product level using standardization and weighting methods, so as to

eliminate the problem of large differences in export product quality results. In terms of the measurement of digital transformation, this paper adopts a text analysis method to extract the relevant statements of digital transformation in the management discussion and analysis (MD&A) part in the annual report, so as to investigate the digital transformation of enterprises. From the perspective of research, considering that digital technology itself has the externality of virtual agglomeration, which can significantly improve the specialized division of labor and collaborative production capacity of products, it is an interesting perspective to consider the promoting effect of technological progress and efficiency improvement on product quality from the perspective of division of labor and collaborative cooperation. From the perspective of enterprises, this paper discusses the influence mechanism of digital transformation on the quality of enterprises' export products and considers the mechanism of innovation performance and total factor productivity, which provides scientific support for in-depth discussion on the path of digital transformation to promote the quality of enterprises' export products. On further analysis, enterprise digital transformation reduces the wage premium ability of low-skilled labor, and even replaces low-skilled employees, widening the internal salary gap of enterprises. The increase of the gap will affect the production efficiency of employees. In this respect, this paper finds that enterprise digital transformation will widen the internal salary gap and, thus, have a masking effect on the quality of export products. The internal mechanism of the internal salary gap on export product quality in digital transformation is discussed.

The rest of this paper is arranged as follows: Section 2 discusses the theoretical mechanism and research hypothesis. Section 3 describes the research design, including model setting, sample selection and data sources. Section 4 provides empirical analysis, including benchmark regression, robustness test, and mechanism analysis. Section 5 is the conclusion and revelation, and discusses the relevant findings.

## 2. Theoretical Mechanism Analysis

The improvement of export product quality needs to be realized through the increase of technical complexity and the improvement of product use value. The characteristics of digital technology are, for example, information transmission across time and space, data creation, sharing and helping to reduce transaction costs, and enabling enterprises to realize the transformation of production mode with the help of digital technology. This paper mainly studies the effect of innovation performance and total factor productivity on the quality improvement of export products in digital transformation.

### 2.1. Innovation Performance

Innovation performance refers to the application of new technologies in production activities by enterprises to improve production efficiency and increase the number of substantive innovations in invention patent applications [27]. Some scholars believe that the number of patent applications of enterprises is a good indicator to measure innovation performance [21], which is different from the number of invention patent applications and non-invention patents included in the index to measure innovation ability. The world has entered the dual-drive era of "innovation" and "data" [28]. At the macro level, digital technology, by virtue of its extensive permeability, data-driven, system intelligence, and other features, reduces R&D costs and improves R&D efficiency, providing more possibilities and development space for export enterprises to carry out innovative R&D and evolve innovation modes [29,30]. Indirectly, it provides the necessary knowledge innovation driving force and efficiency improvement foundation for upgrading the quality of export products. At the meso-level, reusability and replicability of data information promote data elements to be transformed into new knowledge for unlimited reuse among export enterprises in the industry, and the open source and non-competitive characteristics of digital technology determine its extensive diffusion and knowledge spillover effect [31]. Some studies have pointed out that innovation organizations can internalize relevant technical knowledge into new products, services, or processes through absorption capacity,

thus promoting green innovation performance [26]. Therefore, the development of digital economy provides a broader platform and element support for the innovative development of export industry. At the micro level, the application of Internet technology reduces the information asymmetry of the global market, and a fairer international market competition environment makes continuous innovation become the main way for export enterprises to maintain their market position [32]. At the same time, digital technology brings big data, artificial intelligence, blockchain, and other new business forms and models. The large-scale application of digital technology enables enterprises to more accurately grasp the latest consumption preferences in the international market, greatly improve innovation performance, and optimize the enterprise's research and development mode through accurate insight into the massive, personalized consumption demand. It can create a broader space and platform for enterprise innovation, promote the improvement of enterprise innovation performance through digital transformation [33–35], and indirectly promote the quality upgrade of export products, as illustrated in Figure 2. Accordingly, this paper puts forward:

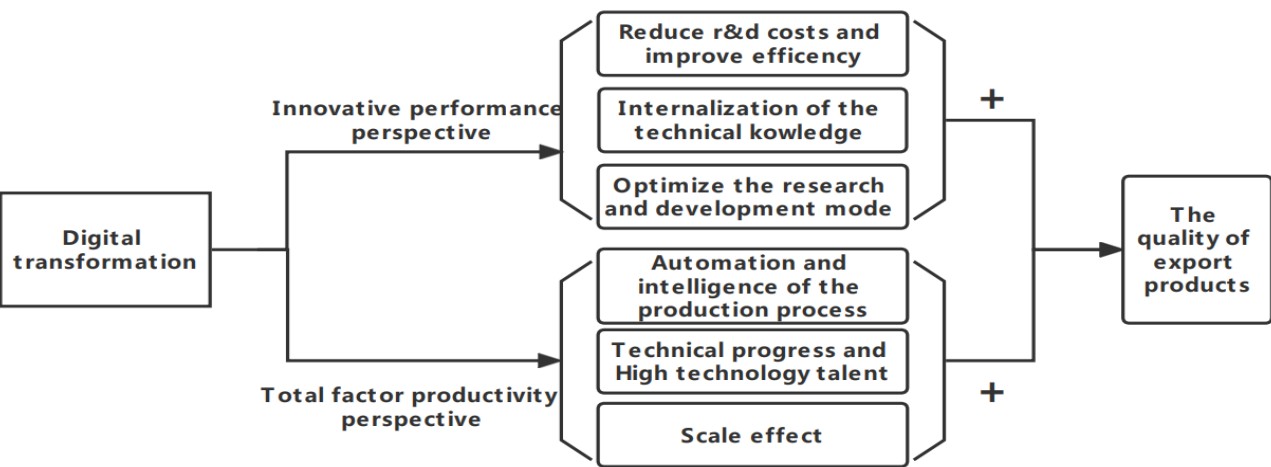

**Figure 2.** The influence mechanism of digital transformation on the quality of export products.

**Hypothesis 1.** *Digital transformation can indirectly promote the quality improvement of export products by improving the innovation performance of enterprises.*

### 2.2. Total Factor Productivity

Total factor productivity refers to the systematic productivity of an enterprise, and the improvement of total factor productivity is the development of industrial upgrading and productivity, including not only the role of technological progress, but also business model innovation, organizational reform, and management mode reform [36]. Firm productivity has an important relationship with the development of international trade. According to Melitz's heterogeneous firm trade theory [6], the impact of firm productivity on export is that only enterprises with higher production efficiency can explore the international market due to higher export costs than in domestic trade. Digital transformation improves total factor productivity by improving enterprise production efficiency [37], promotes technological progress, and realizes scale effect, thus promoting enterprise export. At the same time, free trade reallocates resources from low-productivity enterprises and low-yield products to high-productivity and high-yield products. Under the influence of the externality effect of digital economy, fierce competition in the international market will also promote the digital transformation of enterprises and further improve corporate productivity [38]. First of all, digital transformation can save production cost and improve production efficiency by accelerating automation and intelligence of production process, intelligent control, and precise management. The improvement of enterprise productivity represents the improvement of the enterprise technical level, the digital transformation

of the production process, and the application of intelligent production control system can reduce production loss, improve the level of the manufacturing process, and promote the progress of manufacturing technology, but also directly affect the quality of export products [39,40]. Secondly, in the process of digital transformation, the introduction of digital technology will also train employees and hire high-tech talents to meet the needs of the digital operation of the enterprise, and the highly skilled employees will further improve the production and management level of the enterprise, so as to improve product quality [41]. Finally, the digital transformation of enterprises can innovate production technology at a lower marginal cost, laying the foundation for expanding the scale of manufacturing enterprises. The external effect of digital technology makes enterprises usually choose partners to establish alliances to obtain innovation resources, thus forming the scale effect of the region where the enterprises are located. Total factor productivity of enterprises can be improved through its own scale effect and regional scale effect [42]. It can be seen that digital technology can promote the improvement of total factor productivity in various ways, thus promoting the upgrading of product quality, as illustrated in Figure 2. Accordingly, this paper proposes:

**Hypothesis 2.** *Digital transformation can indirectly promote the quality improvement of export products by improving total factor productivity.*

## 3. Research Design
### 3.1. Model Setting

Step 1: This paper first verifies the impact of enterprise digital transformation on the quality of export products; thus, the model of this paper is set as follows:

$$quality_{ijct} = \alpha_0 + \alpha_1 dig_{it} + vX_{it} + \delta_i + \delta_j + \delta_p + \delta_t + \varepsilon_{ijt} \tag{1}$$

In Formula (1), $quality_{ijct}$ represents the quality of products exported by the enterprise to the country in the year, $dig_{it}$ represents the degree of digital transformation of the enterprise in the year, $\delta_i, \delta_j, \delta_p$ and $\delta_t$ represents the combination of a series of control variables and respectively represents the fixed effects of the enterprise, product, province and city and year, $\varepsilon_{ijt}$ represents the random disturbance term. Let us focus on the coefficients $\alpha_1$.

Step 2: Considering the possible mediating mechanism of enterprise digital transformation on export product quality, according to proposition (1) and proposition (2), we respectively take enterprise innovation performance and enterprise total factor productivity as mediating variables to test the mediating mechanism. The mediation model is as follows:

$$innov_{ijt} = \beta_0 + \beta_1 dig_{it} + vX_{it} + \delta_i + \delta_j + \delta_p + \delta_t + \varepsilon_{ijt} \tag{2}$$

$$tfp_{ijt} = \gamma_0 + \gamma_1 dig_{it} + vX_{it} + \delta_i + \delta_j + \delta_p + \delta_t + \varepsilon_{ijt} \tag{3}$$

The above Equations (2) and (3) mainly focus on coefficient $\beta_1$ and $\gamma_1$, and other variables in the model are consistent with (1), so it will not be repeated here.

### 3.2. Sample Selection and Data Sources

According to the availability of data, this paper takes the database of China Customs enterprises from 2007 to 2015 and China A-share listed companies as the initial samples, and matches customs enterprises with A-share listed companies to form the data of customs listed companies, and carries out the following data screening: (1) ST and ST* enterprises are excluded during the sample investigation; (2) Enterprises whose age is negative during the sample period are excluded; (3) Samples of enterprises with asset-liability ratio greater than 100% are excluded. The data are mainly divided into two parts: the Gutai 'an Database (CSMAR) and the China Customs Enterprise Database. Specific variables are described as follows:

Step 1: For the explained variables, the demand information prediction methods of Khandelwal et al. (2013) [11] and Fan et al. (2015) [8] are used for reference to calculate the export product quality of the 6-digit code (HS6) of customs products. First, construct the regression equation of export product price to export product quality:

$$lnq_{ijct} + \sigma dig_{ijct} = \kappa_i + \kappa_{ct} + v_{ijct} \tag{4}$$

Step 2: In the above Equation (4), $\sigma$ represents elasticity of product substitution, $\kappa_i$ represents the difference between groups of different customs products, $\kappa_{ct}$ controls the fixed effect at the country-year level. The symbols of other variables are consistent with the previous expressions, so we will not repeat them here. The elasticity of product substitution is generally provided by Broda et al. to provide the elasticity value of HS2-bit codes [43]. The residual term $v_{ijct}$ is obtained by regression of the above Equation (4), from which the quality of export products at the level of enterprise-product-exporter-year can be obtained:

$$q_{ijct} = \frac{v_{ijct}}{\sigma - 1} \tag{5}$$

Step 3: Considering the problem that the quality of export products obtained by calculation cannot be compared between different industries of the same enterprise, the quality of export products obtained is standardized:

$$qsd_{ijct} = \frac{q_{ijct} - minq_{ijct}}{maxq_{ijct} - minq_{ijct}} \tag{6}$$

Step 4: Take the ratio of export quantity to the total export of the enterprise as weight, and finally get the quality of export products at the enterprise level:

$$quality_{ijct} = \sum_{j \in \Omega} \frac{quantity_{ijct}}{\sum_c quantity_{ijct}} \times qsd_{ijct} \tag{7}$$

In Formula (7), $\Omega$ represents the product set and $quantity_{ijct}$ represents the quantity of products $f$ exported by the enterprise $i$ to the country $c$ in the year $t$. Other variables are consistent with the previous statements, so we will not repeat them here.

As for the core explanatory variable Digital Transformation Index (dig), firstly, 30 important digital economy policy documents released by the Central People's Government and the Ministry of Industry and Information Technology were collected, and a digital dictionary was constructed to obtain 238 digital keywords. The digital keywords are showen in Appendix A. Secondly, Python software was used to analyze the text of the annual reports of listed companies, the management discussion and analysis (MD&A) part was extracted, and the number of digital keywords appearing in MD&A of each company every year was calculated. Finally, the number of occurrences of each key word in MD&A was added up, and the total number of occurrences of digital key words obtained was divided by the length of MD&A segment to obtain the annual digitalization degree index of each enterprise. The larger the index value, the higher the degree of digital transformation.

As for the intermediary variables, innovation performance (innov) and total factor productivity (tfp) were selected as the intermediary variables of enterprise digital transformation and export product quality according to the hypothesis above. Among them, according to Pasquali [44], innovation performance may sometimes be a strategic behavior to obtain innovation subsidies. Therefore, the quality of invention patent, utility model patent, and design patent is different. Therefore, the weights of the applied invention patents, utility model patents, and design patents were added together, and logarithm was taken as the measurement index of innovation output. The weights of the three kinds of patents were marked with the value of 3:2:1. Second, innovation input was measured by the ratio of R&D expenditure to operating income. Finally, innovation output was divided by innovation input to get innovation performance. For total factor productivity, the LP

method proposed by Levinsohn and Petrin et al. [36] was adopted, because the LP method can effectively solve the problem of sample loss.

For control variables, the following control variables were selected in line with previous literature on the quality of export products of enterprises: (1) Enterprise size (size), logarithm of total enterprise assets; (2) The age of the enterprise (age), the difference between the year of the current year and the year of the establishment of the enterprise is added by 1, and then the logarithm is taken; (3) Return on equity (roe), net profit divided by the average balance of shareholders' equity; (4) Total assets turnover (ato), operating income divided by average total assets; (5) Cashflow ratio (cashflo), net cashflow from operating activities divided by total assets; (6) Asset-liability ratio (lev), total liabilities divided by total assets at the end of the year; (7) Tobin's q (tq), the market value of outstanding shares plus the product of the number of non-tradable shares and the net asset value per share plus the book value of liabilities to the total value divided by the total assets; (8) Herfindahl Index (hhi), using the Herfindahl index to calculate the industry monopoly index of enterprises, the calculation formula is $hhi = \sum_{i=1}^{n}\left(total\_sale_{ij}\right)/\sum_{i=1}^{n} total\_sale_{ij})^2$, where, $total\_sale_{ij}$ represents the main business income of enterprises in the industry. Table 1 below reports descriptive statistics of the above variables and Table 2 reports variable definition.

**Table 1.** Descriptive statistics.

| Variable Type | Variable | Variable Name | Sample Size | Mean | Standard Deviation | Minimum | Maximum |
|---|---|---|---|---|---|---|---|
| Explained variable | quality | Quality of export products | 410,325 | 0.0325 | 0.1180 | 0.0000 | 1.0000 |
| Core explanatory variable | dig | Digital transformation | 410,325 | 0.0059 | 0.0063 | 0.0000 | 0.0521 |
| Mediating variable | innov | Innovation performance | 346,326 | 0.2237 | 0.0903 | 0.0000 | 0.4315 |
| | tfp | Total factor productivity | 410,325 | 8.5174 | 1.1587 | 5.2837 | 11.6660 |
| Control variable | size | Enterprise size | 410,325 | 22.3726 | 1.5302 | 17.8787 | 27.7033 |
| | age | The age of the enterprise | 410,325 | 0.9459 | 0.1754 | 0.0940 | 1.3589 |
| | roe | Return on equity | 410,325 | 0.0998 | 0.1331 | −2.2918 | 0.7108 |
| | ato | Total assets turnover | 410,325 | 0.8875 | 0.5337 | 0.0243 | 7.8714 |
| | cashflow | Cashflow ratio | 410,325 | 0.0497 | 0.0747 | −1.9377 | 0.4300 |
| | lev | Asset-liability ratio | 410,325 | 0.4636 | 0.2095 | 0.0075 | 0.9970 |
| | tq | Tobin's q | 402,660 | 1.9589 | 1.1847 | 0.7488 | 48.5054 |
| | hhi | Herfindahl Index | 402,470 | 0.2086 | 0.2633 | 0.0131 | 1.0000 |

**Table 2.** Sample variable definition and description.

| Variable Name | Variable | Variable Definition |
|---|---|---|
| Quality of export products | quality | Firstly, the export product quality is calculated according to the demand information prediction method, and then standardized. Finally, the export product quality is calculated according to the weight of the export quantity. |
| Digital transformation | dig | Using the text analysis method of machine learning, the keyword frequency of enterprise annual report about digital transformation is calculated as the measurement index of enterprise digital transformation. |
| Innovation performance | innov | According to different patent types, invention patents, utility model patents, and design patents are weighted at 3:2:1, and divided by the ratio of R&D expenses to operating income. |
| Total factor productivity | tfp | Calculated according to the LP method. |
| Enterprise size | size | Logarithm of total enterprise assets. |
| The age of the enterprise | age | The logarithm of the difference between the current year and the year of establishment is added by one. |
| Return on equity | roe | Net profit divided by the average balance of shareholders' equity. |
| Total assets turnover | ato | Operating income divided by average total assets. |
| Cashflow ratio | cashflo | Net cash flow from operating activities divided by total assets. |
| Asset-liability ratio | lev | Total liabilities at year-end divided by total assets at year-end. |
| Tobin's q | tq | Market value of outstanding shares plus the number of non-tradable shares multiplied by net asset value per share plus the book value of liabilities to total value divided by total assets. |
| Herfindahl Index | hhi | $hhi = \sum_{i=1}^{n}\left(total\_sale_{ij}\right)/\sum_{i=1}^{n} total\_sale_{ij})^2$ |

## 4. Empirical Analysis

### 4.1. Baseline Regression Analysis

Table 3 below shows the baseline regression of the impact of enterprise digital transformation on export product quality. Among them, (1) is listed as the result of fixed effect of controlling enterprise, year, and export products. It can be seen that the dig coefficient is 0.1337, which is significantly positive at 1% level, indicating that enterprise digital transformation can significantly improve the quality of its export products. Furthermore, (2) is listed as other characteristics of control enterprises, and it is found that the significance and direction of the dig regression coefficient do not change, and it is still significantly positive at 1% level. In columns (3) and (4), the fixed effect of provinces and cities and other firm control variables were added, and the significance and direction of the dig coefficient remained unchanged and passed the 1% significance level test. The results show that the digital transformation of Chinese enterprises can significantly promote the quality of their export products. It can be seen that under the tide of digital revolution, enterprise digital transformation can provide the necessary knowledge and efficiency improvement basis for enterprise production, apply it to the production link, accurately analyze consumer preferences through big data, and improve product quality while reducing the cost of information search.

**Table 3.** Basic regression result.

| Variable | (1) | (2) | (3) | (4) |
|---|---|---|---|---|
| dig | 0.1337 *** | 0.1932 *** | 0.1965 *** | 0.2048 *** |
| | (0.0489) | (0.0499) | (0.0501) | (0.0512) |
| size | - | −0.0035 *** | −0.0038 *** | −0.0032 *** |
| | | (0.0008) | (0.0009) | (0.0010) |
| age | - | −0.0196 *** | −0.0203 *** | −0.0203 *** |
| | | (0.0053) | (0.0054) | (0.0055) |
| roe | - | 0.0054 ** | 0.0053 * | 0.0030 |
| | | (0.0025) | (0.0028) | (0.0028) |
| ato | - | - | −0.0005 | 0.0006 |
| | | | (0.0013) | (0.0014) |
| cashflow | - | - | 0.0097 ** | 0.0075 * |
| | | | (0.0040) | (0.0041) |
| lev | - | - | 0.0044 | 0.0033 |
| | | | (0.0032) | (0.0033) |
| tq | - | - | - | 0.0003 |
| | | | | (0.0002) |
| hhi | - | - | - | −0.0031 *** |
| | | | | (0.0011) |
| cons | 0.0316 *** | 0.1290 *** | 0.1337 *** | 0.1204 *** |
| | (0.0003) | (0.0200) | (0.0216) | (0.0228) |
| Firm fixed effect | Yes | Yes | Yes | Yes |
| Year fixed effect | Yes | Yes | Yes | Yes |
| Product fixed effect | Yes | Yes | Yes | Yes |
| Region fixed effect | No | Yes | Yes | Yes |
| Observations | 410,325 | 410,325 | 410,325 | 394,814 |
| R-squared | 0.0976 | 0.0977 | 0.0978 | 0.1021 |

Note: Small brackets () are robust standard error, *, ** and *** respectively indicate significant at the level of 10%, 5%, and 1%, the same below.

### 4.2. Robustnesstest

4.2.1. Instrumental Variable Regression

This paper mainly verifies the impact of digital transformation on the quality of export products, but this inspection mechanism may have endogenous problems. Although the empirical results show that enterprise digital transformation may improve the quality of

export products, enterprises with high quality export products have stronger willingness or capital to adopt digital technology to promote enterprise digital transformation, which leads to the deviation of reverse causation. In order to alleviate the endogeneity problem, the two-stage least square method (2SLS) was used in this paper to re-estimate, so as to reduce the deviation of the endogeneity problem in the research results.

Firstly, the average level of digital transformation of enterprises in the same industry in the region is selected as the instrumental variable. In theory, the digital transformation level of the same industry in the region where the enterprise is located can promote the adoption of digital technology and meet the conditions of correlation with the core explanatory variables. However, the degree of digital transformation in the same industry in the same region does not directly affect the quality of export products of enterprises, which meets the exogenous conditions. Secondly, the number of Internet access ports in the province where the enterprise is located is selected as the instrumental variable of digital transformation. The number of Internet access ports reflects the degree of residents' participation on the Internet in each province. The higher the degree of Internet participation is, the more advanced the digital technology is in the province. The number of Internet access ports does not directly affect the quality of enterprises' export products but can stimulate enterprises to carry out industrial digital empowerment, improve the level of digital transformation, and satisfy the exogeneity hypothesis of instrumental variables. Finally, the degree of digital transformation of all enterprises in the province where the enterprise is located is added up to construct instrumental variables for the overall digital transformation of the province where the enterprise is located [45]. Regression of instrumental variables is shown in Table 4 below. It can be seen that the regression results of the three instrumental variables all showed positive dig coefficients and passed the significance level test of more than 5%. In addition, the statistics of Kleibergen–Paap rk LM and the critical value of Kleibergen–Paap rk Wald F both reject the null hypothesis, indicating that the instrumental variables reject the unidentifiable and weakly identifiable tests, respectively, and the selection of instrumental variables is reasonable.

**Table 4.** Instrumental variable regression.

| Variable | (1) | (2) | (3) |
|---|---|---|---|
| Dig | 1.9732 *** | 1.9707 *** | 0.6367 ** |
| | (0.4112) | (0.4591) | (0.2851) |
| Kleibergen–Paap rk LM | 1623.145 | 4196.642 | $1.1 \times 10^4$ |
| | [0.0000] | [0.0000] | [0.0000] |
| Kleibergen–Paap rk Wald F | 2842.891 | 4583.238 | $1.1 \times 10^4$ |
| | {16.38} | {16.38} | {16.38} |
| Control variable | Yes | Yes | Yes |
| Firm fixed effect | Yes | Yes | Yes |
| Year fixed effect | Yes | Yes | Yes |
| Product fixed effect | Yes | Yes | Yes |
| Region fixed effect | Yes | Yes | Yes |
| Observations | 394,814 | 394,814 | 394,814 |

Note: ** and *** respectively indicate significant at the level of 5%, and 1%. Brackets [] are chi-square *p*-values used to test whether instrumental variables are unidentifiable, and curly brackets {} represent 10% critical values used to identify weakly identifiable instrumental variable statistics, the same below.

In this paper, core variables were replaced and then returned to test the rationality of the model construction. First, precise word frequency interception is no longer used for core explanatory variables, so as to expand the length of MD&A and construct new digital transformation indicators. Secondly, the quality of export products is not weighted and standardized export product quality indicators are directly used. Finally, change the weighting method of export product quality and adopt the weighting method of trade amount instead of trade volume. The results are shown in Table 5. It can be seen that the dig coefficient is still significantly positive after replacing the core variable.

**Table 5.** Replacement variable regression.

| Variable | Replace the Explanatory Variable | Replace the Explained Variable | |
|---|---|---|---|
| | **(1)** | **(2)** | **(3)** |
| Dig | 0.0020 *** | 0.3469 *** | 0.1610 *** |
| | (0.0005) | (0.1157) | (0.0474) |
| Control variable | Yes | Yes | Yes |
| Cons | 0.1205 *** | 0.3621 *** | 0.1165 *** |
| | (0.0229) | (0.0469) | (0.0216) |
| Firm fixed effect | Yes | Yes | Yes |
| Year fixed effect | Yes | Yes | Yes |
| Product fixed effect | Yes | Yes | Yes |
| Region fixed effect | Yes | Yes | Yes |
| Observations | 394,814 | 394,814 | 394,814 |
| R-squared | 0.1021 | 0.0482 | 0.1093 |

*** respectively indicate significant at the level of 1%.

### 4.2.2. Other Robustness Tests

A series of other robustness tests were also conducted in this paper. First, core variables of the samples were treated with 1% tail reduction to avoid bias of the regression results caused by outliers. Secondly, the financial crisis had a great impact on global trade, which also caused bias to the regression estimate, so the sample data of 2008 and 2009 were excluded. Finally, since processing trade is a re-export business activity, which cannot fully reflect the product quality of enterprises, this paper only retains general trade samples and excludes other trade mode samples, so as to better estimate the impact of enterprises' digital transformation on the quality of export products. The results are shown in Table 6 below. After the above robustness test, the dig coefficient is still significantly positive, indicating that the model construction is reasonable.

**Table 6.** Other robustness tests.

| Variable | Tail Reduction Treatment | Strip Out the Effects of the Financial Crisis | Only Keep the General Mode of Trade |
|---|---|---|---|
| | **(1)** | **(2)** | **(3)** |
| Dig | 0.2048 *** | 0.3087 *** | 0.1284 ** |
| | (0.0513) | (0.1183) | (0.0532) |
| Control variable | Yes | Yes | Yes |
| Cons | 0.1207 *** | 0.3699 *** | 0.1139 *** |
| | (0.0229) | (0.0501) | (0.0238) |
| Firm fixed effect | Yes | Yes | Yes |
| Year fixed effect | Yes | Yes | Yes |
| Product fixed effect | Yes | Yes | Yes |
| Region fixed effect | Yes | Yes | Yes |
| Observations | 394,814 | 371,380 | 326,268 |
| R-squared | 0.1000 | 0.0476 | 0.1118 |

** and *** respectively indicate significant at the level of 5%, and 1%.

### 4.3. Inspection of Channels and Mechanisms

In the previous part, we verified that the digital transformation of enterprises has an obvious promoting effect on the quality of their export products, and this part verifies the channels and mechanisms. According to the above theoretical analysis, the digital transformation of enterprises provides more innovation modes to improve the innovation efficiency of enterprises, and digital technology optimizes the production mode and management mode of enterprises, improves the total factor productivity, and, thus, improves the quality of enterprises' export products. Next, based on the above logic, we tested the following two questions to demonstrate the impact mechanism of enterprise digital transformation

and export product quality: First, whether enterprise digital transformation can indirectly promote the improvement of export product quality by improving enterprise innovation performance; Second, whether enterprises' digital transformation can indirectly promote the quality improvement of export products by improving total factor productivity. The results are shown in Table 7 below.

**Table 7.** Inspection of channels and mechanisms.

| Variable | Innovation Performance | Total Factor Productivity |
|---|---|---|
| | **(1)** | **(2)** |
| Dig | 11.1893 *** | 14.5559 *** |
| | (0.6585) | (0.5099) |
| Kleibergen–Paap rk LM | 770.931 | 1623.145 |
| Kleibergen–Paap rk Wald F | 506.725 | 2842.942 |
| Control variable | Yes | Yes |
| Firm fixed effect | Yes | Yes |
| Year fixed effect | Yes | Yes |
| Product fixed effect | Yes | Yes |
| Region fixed effect | Yes | Yes |
| Observations | 337,185 | 394,724 |

*** respectively indicate significant at the level of 1%.

### 4.3.1. Innovation Performance Mechanism

According to the theoretical analysis above, enterprise digital transformation can form an open innovation, improve the possibility of innovation cooperation, effectively reduce the cost of research and development, so as to improve the efficiency of research and development, make better use of innovation resources, improve the technical level of products, and help enterprises to improve the quality of export products. Iv-2sls was used to empirically test the impact of digital transformation on innovation performance, and the average level of digital transformation in the same industry was selected as the instrumental variable. The results are shown in Column (1). The results show that enterprise digital transformation can significantly improve innovation performance and pass the 1% significance level test [46]. Therefore, innovation performance is verified in the intermediary mechanism between enterprise digital transformation and export product quality. That is, the application of digital technology can integrate innovative resources. On the one hand, it can improve the enterprise's data acquisition ability and reflect the sales situation of new products faster. On the other hand, it improves the enterprise data analysis ability and can reflect the market demand more accurately. The application of digital technology promotes the close combination of R&D end and consumption end and improves the efficiency of product R&D. In addition, digital technology has the positive externality of virtual agglomeration, which can improve the collaborative R&D efficiency among enterprises, break through the innovation bottleneck of enterprises, develop new products, and improve the export competitiveness of enterprises.

### 4.3.2. Total Factor Productivity Mechanism

According to the theoretical analysis above, digital transformation of enterprises can improve the total factor productivity of enterprises by improving the management mode of enterprises, enhancing the intelligent production and manufacturing level of enterprises and expanding the scale economy effect of enterprises, and the improvement of total factor productivity of enterprises is conducive to reducing the production and operating costs of enterprises, so that enterprises have more costs to improve products, develop new products, and improve the quality of export products. Similarly, IV-2SLS was used to empirically test the impact of digital transformation on total factor productivity of enterprises, and the average level of digital transformation of enterprises in the same industry was selected as instrumental variable. The results are shown in Column (2). The results show that an

enterprise's digital transformation can significantly improve the total factor productivity and pass the 1% significance level test. Therefore, the mediating mechanism of total factor productivity in enterprise digital transformation and export product quality was verified. In other words, enterprises automate routine tasks through digital transformation, simplify operating costs of enterprises, improve marketing and production mode, and enhance the collaborative operation of departments, which is conducive to the formation of new production power. In addition, the digital data information collection and processing capability can make the relationship between enterprises and upstream and downstream enterprises closer, improve the operation efficiency of the industrial chain, and, thus, enhance the total factor productivity of enterprises. The improvement of total factor productivity of enterprises means the reduction of production cost. Low production cost is conducive to setting lower prices for enterprises, and the quality of products of enterprises with high productivity will also be improved.

### 4.4. Heterogeneity Analysis

This part was put into heterogeneity analysis, and the results are shown in Table 8 below. The heterogeneity analysis mainly includes three aspects: 1. Based on the heterogeneity analysis of enterprise ownership, enterprises can be divided into foreign-funded enterprises and domestic-funded enterprises according to their ownership. Foreign-funded enterprises and domestic-funded enterprises have different driving forces in digital transformation, which will have different impacts on the quality of export products. Therefore, the virtual variable owner was constructed, and the assignment method is 1 and 0; that is, if it is a foreign-funded enterprise, the assignment is 1, otherwise it is 0, and the assignment is the same for a domestic enterprise. Columns (1) and (2), respectively, show the regression results of foreign-funded enterprises and domestic enterprises. It can be seen that the digitalized transformation of domestic enterprises has stronger export product quality improvement effect than that of foreign-funded enterprises, which may be because most foreign-funded enterprises are backward production capacity enterprises in developed countries and their industrial technology level is not high, thus affecting the digitalized transformation's effect on the quality improvement of export products. 2. Based on regional heterogeneity analysis, the provinces and cities where the enterprises are located were divided into eastern region and central and western region (eastern region: Beijing, Tianjin, Hebei, Shandong, Shanghai, Jiangsu, Zhejiang, Fujian, Guangdong, Hainan, etc. Due to the selection of A-share listed companies, Hong Kong, Macao, and Taiwan regions are not taken into account, and other provinces and cities are central and western regions.) The assignment method also assigns values of 1 and 0. China's regional development presents a cascade distribution in the east and west, so the development degree of digital economy will also be different, and the quality effect of export products of enterprises' digital transformation will have regional heterogeneity. Columns (3) and (4), respectively, show the heterogeneity results of the eastern and central regions. It can be seen that compared with the central and western regions, the digital transformation of enterprises in the eastern region has a more obvious effect on improving the quality of export products. 3. Based on technical heterogeneity analysis, enterprises were divided into technology-intensive and non-technology-intensive enterprises according to their technology-intensive degree. Technology-intensive enterprises are more motivated to adopt digital technology, which will affect the quality of export products. The technology-intensive classification standard is the median ratio of R&D personnel to the total number of employees, which is calculated as 0.1156. If the value exceeds this value, it indicates that the enterprise is technology-intensive, and the assignment method is 1 and 0. From columns (5) and (6), it can be seen that the digitalized transformation of technology-intensive enterprises has a more obvious effect on improving the quality of export products.

**Table 8.** Heterogeneity analysis.

| Variable | Ownership Heterogeneity | | Regional Heterogeneity | | Technical Heterogeneity | |
|---|---|---|---|---|---|---|
| | Foreign Capital | Domestic Capital | The Eastern Region | The Central and Western Regions | Technology Intensive | Non-Technology-Intensive |
| | (1) | (2) | (3) | (4) | (5) | (6) |
| Dig | 0.2022 ** | 0.2386*** | 0.2578 *** | 0.0344 | 3.2488 ** | −1.7769 |
| | (0.0948) | (0.0607) | (0.0575) | (0.1226) | (1.3926) | (2.6642) |
| Control variable | Yes | Yes | Yes | Yes | Yes | Yes |
| Cons | 0.2065 ** | 0.1149 *** | 0.1357 *** | 0.0866 ** | −2.9151 *** | 1.1393 |
| | (0.0801) | (0.0244) | (0.0278) | (0.0416) | (1.0264) | (0.7977) |
| Firm fixed effect | Yes | Yes | Yes | Yes | Yes | Yes |
| Year fixed effect | Yes | Yes | Yes | Yes | Yes | Yes |
| Product fixed effect | Yes | Yes | Yes | Yes | Yes | Yes |
| Region fixed effect | Yes | Yes | Yes | Yes | Yes | Yes |
| Observations | 64,685 | 330,129 | 312,149 | 82,665 | 34,732 | 34,716 |
| R-squared | 0.1672 | 0.0991 | 0.0858 | 0.2128 | 0.2619 | 0.1698 |

** and *** respectively indicate significant at the level of 5%, and 1%.

### 4.5. Further Analysis: The Masking Effect of Income Share

Research shows that digital transformation will increase the demand for high-skilled labor, while conventional low-skilled labor will be replaced by artificial intelligence [47]. As an applied technology, digital transformation itself has an obvious "skill-based technological progress" effect [48], thus enhancing the wage bargaining power of unconventional labor, and, thus, widening the salary gap between top and bottom employees of enterprises. The widening of the salary gap within an enterprise is likely to cause discontent among employees at the bottom, thus affecting work efficiency [49]. Such reduction of work efficiency may have a negative impact on the quality of enterprise products, and then affect the quality of export products. Therefore, there is a reverse effect between the intermediary mechanism and the direct effect mechanism, that is, the uneven share of enterprise income will have a masking effect on the quality upgrade effect of enterprise digital transformation.

Step 1: We explored the mechanism effect of changes in internal revenue share on digital transformation and export product quality. The internal income share of enterprises was investigated from two perspectives. On the one hand, the internal salary gap (lnequapay) was used as the index of the internal salary gap, and the ratio of the average salary of management and the average salary of employees was taken as logarithm. On the other hand, there is the average salary level of ordinary employees (lnwage), which was measured by dividing cash paid to and for employees by the number of employees and taking logarithm.

Step 2: To verify the above mechanism, the following regression model was constructed:

$$lnequapay_{ijt} = \alpha_0 + \alpha_1 dig_{it} + vX_{it} + \delta_i + \delta_j + \delta_p + \delta_t + \varepsilon_{ijt} \tag{8}$$

$$lnwage_{ijt} = \gamma_0 + \gamma_1 dig_{it} + vX_{it} + \delta_i + \delta_j + \delta_p + \delta_t + \varepsilon_{ijt} \tag{9}$$

Step 3: IV—2SLS was adopted for estimation, and the above model was mainly concerned with the coefficients. The results are shown in Table 9 below. As can be seen from Column (1), the application of digital technology will expand the internal salary gap and reduce the average salary level of enterprise employees, indicating that the application of digital technology will eliminate part of regular task employees, reduce the wage premium ability of low and medium skills, and then increase the internal salary gap and reduce the average salary level of employees. The widening of the internal salary gap and the reduction of the salary level will affect the production enthusiasm of employees, and then reduce the production efficiency, which has a negative effect on the product quality of the enterprise. It can be seen that although digital transformation can promote the quality of

enterprises' export products on the whole, the application of digital technology also brings uneven income distribution of enterprises, which has a negative "masking effect" on the quality of export products.

**Table 9.** Further analysis: the masking effect of income share.

| Variable | Intra-Firm Compensation Dispersion | Average Salary Level of Enterprise Employees |
|---|---|---|
| | (1) | (2) |
| Dig | 46.7025 *** <br> (1.6504) | −43.5994 *** <br> (1.5099) |
| lnequapay | - | - |
| lnwage | - | - |
| Kleibergen–Paap rk LM | 1620.142 <br> [0.0000] | 1620.142 <br> [0.0000] |
| Kleibergen–Paap rk Wald F | 2837.910 <br> {16.38} | 2837.910 <br> {16.38} |
| Control variable | Yes | Yes |
| Firm fixed effect | Yes | Yes |
| Year fixed effect | Yes | Yes |
| Product fixed effect | Yes | Yes |
| Region fixed effect | Yes | Yes |
| Observations | 393,479 | 393,479 |

*** respectively indicate significant at the level of 1%.

## 5. Empirical Results

As an important driving mode of economic development, digital transformation has been an important path for countries to move towards industry 4.0. In particular, coastal areas such as Guangdong, which have the advantages of economy, geography, and resources, have always been in the forefront of digital transformation. For a long time, the economy of the Guangdong Province has been developing rapidly and with high quality, and the contribution of small- and medium-sized enterprises cannot be ignored. The number of small- and medium-sized manufacturing enterprises in the Guangdong Province accounts for 95% of the total number of manufacturing enterprises. It can be seen that small- and medium-sized enterprises are not only the main body of digital transformation, but also the focus and difficulty of transformation. In China's 14th five-year plan, the development goal of the digital economy is to move towards a comprehensive expansion period by 2025 and a prosperous and mature period by 2035. This means that small- and medium-sized manufacturing enterprises must implement digital transformation under the new national economic development requirements. For traditional manufacturing enterprises, they must try to start with the purchase of digital equipment and gradually realize intelligent manufacturing. For emerging technology enterprises with inherent advantages, they will choose to carry out digital reform in terms of organizational reform and talent skills training.

However, due to differences with large enterprises in various aspects, these small- and medium-sized manufacturing enterprises have encountered great difficulties in the process of digital transformation. Under the pressure caused by the general environment of industrial digital transformation, most enterprises have not clearly defined the digital transformation objectives, do not understand their own preparations, and still less do they know the methods and steps of digital implementation. In addition, the enterprises have limited resources and poor risk resistance, so they either dare not implement digitization at will or may blindly reform. Wrong decisions can waste resources. In addition, digital transformation is system engineering, involving every aspect of the enterprise. Enterprises cannot evaluate the relationship between these standards. In this study, DEMATEL-ANP was used to simplify the complex relationship between these standards by providing the structure of digital maturity and determine the importance of each standard. Then, the fuzzy comprehensive evaluation method was used to evaluate and compare the typical

industries, and the relationship between the indicators of digital maturity was analyzed in depth to provide effective guidance for the digital transformation of small- and medium-sized manufacturing enterprises.

## 6. Conclusions and Enlightenment

### 6.1. Research Conclusions

This paper focuses on the theoretical mechanism and action mechanism of digital transformation on the improvement of export product quality of manufacturing enterprises and uses the export data of foreign trade enterprises and the annual report data of listed foreign trade enterprises from 2007 to 2015 to measure the export product quality and enterprise digital transformation index, and empirically tests the impact of digital transformation on the quality of export products. Digital transformation comes from the development of digital technology, which also promotes the specialized division of labor of the real estate industry and the collaborative production capacity of the industrial chain. It is found that enterprise digital transformation can significantly improve the quality of enterprise export products. This is because digital technology has a virtual agglomeration externality, through which enterprise R&D performance and production efficiency can be improved. Therefore, there are two mechanisms, innovation efficiency and total factor productivity, to improve the quality of export products. This also verifies that the specialization and inter-enterprise or inter-departmental collaboration brought by digital transformation are the points worth paying attention to in current enterprises. In the heterogeneity analysis, the quality improvement effect of export products in digital transformation of domestic enterprises is greater than that of foreign enterprises, and the quality improvement effect of export products in digital transformation of technology-intensive enterprises in eastern regions is more significant. Furthermore, the digital transformation of enterprises will significantly expand the salary gap among employees, which may have an adverse impact on the quality of export products. In other words, there is a masking effect, and it also reflects that the initial phase of digital transformation will expand the income inequality of employees.

### 6.2. Policy Inspiration

6.2.1. Digital-Related Technologies Are Deeply Integrated into Every Link of Manufacturing Enterprises

It is important to be fully aware of the current development status of Chinese manufacturing industry and the new development opportunities brought by digital transformation for enterprises, to vigorously support the digital transformation of manufacturing enterprises, truly integrate advanced digital technology into enterprise business model innovation, including the optimization and upgrading of research and development mode, procurement mode, production mode, and sales mode, improve the specialization and collaborative research, and to develop the efficiency of enterprises. To accelerate the innovation and research of new products of enterprises, improve the quality of export products of foreign trade enterprises in an all-round way, so that China's manufacturing enterprises have the ability to develop continuously.

6.2.2. The Transformation of Digital Transformation Thinking

The difficulties of digital transformation of manufacturing enterprises include a long cycle, slow effect, high transformation investment, and poor transformation effect. The reason is that manufacturing enterprises should change their understanding of digital transformation from tool transformation thinking to "real digital enterprise" thinking. Digital transformation is not a technical problem at the root, but a transformation of the overall strategy of the organization. The digital transformation of enterprises should have a strong desire for transformation, formulate transformation strategies and objectives, select the scene, and implement the distribution, dare to trial and error, and unswervingly invest resources.

### 6.2.3. Establish an Open Innovation ecosystem for Digital Technology Industry and Manufacturing Application Scenarios

We should actively promote the construction of industrial integration and innovation of digital technology industry and manufacturing industry. Digital technology enterprises, R&D personnel of digital technology platform enterprises, and technical personnel of manufacturing enterprises should jointly develop enterprise digital platform technology and digital transformation solutions, so as to promote the implementation of digital transformation. By providing an open innovation platform, traditional manufacturing enterprises establish an open innovation ecosystem around their own products, and digital technology research and development enterprises provide original ecological technology supplies to attract digital technology developers to conduct technology research and development of new functions within the ecosystem, and accelerate digital upgrading through technology internalization and application.

Of course, there are still some limitations in this paper: Limited by data, this paper only investigates the enhancement effect of digital transformation on export product quality at the enterprise level but cannot track the dynamic influence of digital transformation on new product changes and product transformation. In addition, the data are from 2007 to 2015, and in recent years, digital economy has developed more rapidly, and its promoting effect on the quality of export products may have new characteristics. Obviously, to comprehensively assess the long-term impact of enterprise digital transformation on the quality of export products, a wider range of empirical data and a longer period of time are needed, which is also one of the directions for further research in this field.

**Author Contributions:** Conceptualization, F.W. and L.Y.; Data curation, L.Y. and F.W.; Formal analysis, L.Y.; Methodology, L.Y. and F.W.; Project administration, F.W. and L.Y.; Resources, Software L.Y.; Supervision, F.W. and L.Y.; Validation, L.Y. Writing—original draft, F.W. and L.Y.; Writing—review & editing, F.W. and L.Y. All authors have read and agreed to the published version of the manuscript.

**Funding:** This research was funded by the following four projects. Higher Education Science Special Project of Guangdong Provincial Education Department in 2022, "Entrepreneurship Incubation, Collaborative Promotion, Construction of Mass Entrepreneurship Training System—Based on cross-border E-commerce Internship Platform for school-enterprise Cooperation" (project number: 2022GXJK385). "Crisis Response Mechanism and Policy Research of Small- and Medium-sized Enterprises in Guangdong in the Post-COVID-19 Era" (project number: 2021ZDJS141). "The Guangdong Provincial Social Science Planning 2022 Regular General Project", "Research on the Spatial Allocation of Labor Force in Guangdong Province under the Background of Industrial Intelligence" (project number: GD22CLJ01). It was supported by the Construction Project of Public Management, a key discipline with Characteristics of Guangdong Province in 2016 (project number: 2017STSZD01).

**Institutional Review Board Statement:** Not applicable.

**Informed Consent Statement:** Not applicable.

**Data Availability Statement:** The original contributions presented in this study are included in the article; further inquiries can be directed to the corresponding author.

**Acknowledgments:** Wanling Chen and of Guangdong University of science and technology for their constructive guidance.

**Conflicts of Interest:** The authors declare no conflict of interest.

## Appendix A

**Table A1.** Digital transformation index.

| Keywords of Digital Transformation |
| --- |
| Information; networking; data; Internet; intelligence; informatization; artificial intelligence; digitalization; intelligent key technologies; information technology; e-commerce; communications; core technologies; industrial chain; virtual reality; networking; broadband; machines; information security; information systems; data centers; connectivity; |
| cyberspace; industry-university-research; human-machine; interaction; data sharing; data security; number Data base; sensor; e-government; data analysis; wireless; network; e-commerce; Internet security; information network; integrated circuit; information network; public data; technology development; software and hardware; information industry; radio and television; radio and television; technology transformation; numerical control; energy network; network coverage; electric; algorithm; communication network; cross-media; computer; gateway; automation; television network; Service network; data service; data flow; application software; service network; data processing; data mining; digital television; network facilities; broadband access; data management; information management; online education; server; computing technology; automatic control; processors; development tools; control technology; network services; network equipment; product development; electronic information; invention patents; high Technology; high and new technology; monitoring network; portal network; portal website; live broadcast; smart phone; intelligent network; networking; navigation system; multimedia; Internet protocol; base station; agricultural remote sensing; human-computer interaction; satellite communication; radio; wireless network; wireless network; information port; domain name; terminal products; bit; coding; electronic products; management information system; national defense technology; communication satellite; information flow; virtualization; All Access; government network; intelligent algorithm; China Association for Science and Technology; Business intelligence; image understanding; investment decision aid system; intelligent data analysis; intelligent robotics; machine learning; deep learning; semantic search bio metrics; face recognition; speech recognition; authentication; automatic driving; natural language processing; big data; text mining; data visualization Integration; heterogeneous data; credit information; augmented reality; mixed reality; block-chain; digital currency; distributed computing; differential privacy technology; intelligent financial contracts; cloud computing; stream computing; graph computing; memory computing; cognitive computing; fusion architecture;100 million level concurrency; Internet of Things; information physical system; mobile Internet; industrial Internet; mobile Internet; Internet medical; mobile payment; Third party payment; intelligent energy; Internet connection; intelligent wearable; intelligent agriculture; intelligent transportation; intelligent medical care; intelligent customer service; intelligent home; intelligent investment; intelligent cultural travel; intelligent environmental protection; intelligent power grid; intelligent marketing; digital marketing unmanned retail; Internet finance; digital finance; financial technology; quantitative finance; open banking; digital technology; application data; digital; digital number Data management; data network; data platform; data science; digital control; digital communication; digital network; digital intelligence; digital terminal; cloud ecology; cloud service; cloud platform; e-commerce mobile Internet; industrial Internet; Internet solutions; Internet technology; Internet thinking; Internet action; Internet business; Internet mobile; Internet application; Internet marketing; Internet strategy; Internet platform; Internet model; Internet business model; Internet ecology; e-commerce mobile Internet; machine learning Internet business model; cloud storage; Internet+;relational database; blockchain; business intelligence; Business intelligence; industry 4.0. Platform economy; digital creativity; digital business; digital technology; data empowerment; new industrialization; intelligent manufacturing; intelligent technology; intelligent terminal; robotics; ecological collaboration; knowledge management; online; network security; network retail; multi-party security computing; brain-like computing; green computing digital supply chain; intelligent supply chain; supply chain. |

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
