# Peer review of "Digital Transformation and Export Quality of Chinese Products: An Analysis Based on Innovation Efficiency and Total Factor Productivity"

_sustainability, doi:10.3390/su15065395_

Round 1

Reviewer 1 Report

This paper uses the export data of foreign trade enterprises and the annual report data of listed foreign trade enterprises from 2007 to 2015 to measure the export product quality and enterprise digital transformation index, and empirically tests the impact of digital transformation on the quality of export products. It is found that enterprise digital transformation can significantly improve the quality of export products.

The theoretical significance and the practical significance are quite outstanding.The structure of the thesis is reasonable and the expression is clear. Agreed to publish。

Author Response

The paper has been revised, please see the attachment.

Reviewer 2 Report

This paper aims to examine the impact of digital transformation of Chinese listed companies on the quality of their export products. I careful read this paper, I think this paper is interesting and well organized. However, this paper should conduct a major revision before accept. Some suggestions are given as follows:

1.     The contributions of this paper are not clearly, authors should clearly give the contributions in the first section.

2.     Literature Review: I suggest authors cite some paper in the recent years, you can refer to Liu at al 2021 Opening the Box of Subsidies: Which is More Effective for Innovation?.

3.     Theoretical mechanism analysis: Some paragraphs donot cite any references to support the ideas, I suggest cite some paper in the recent years, you can refer to Tian XL. 2020. Geographic Distance, Venture Capital and Technological Performance: Evidence from Chinese Enterprises.

4.     The introduction does not well written, authors should give the background, reason and main contributions, etc.

5.     In the empirical section, the authors should give more explanations for the empirical results instead of reporting the results

6.     Authors need to check the full text.

Author Response

Thank you very much for your valuable review comments. We have revised them according to your comments, please see the attachment

Reviewer 3 Report

The article is very interesting, well-written and well supported. The statistical analysis is thorough, compete and exhaustive. The only recommendation is to avoid significance levels of 10%.

For a better understanding of section 4 , authors are required to provide a table explaining the meaning of each variable involved in the models, their range of values and the scale of measurement used.

The following errors were also detected:

•             Figure 1. “2. Diglitalization”

•             Lines 111, 120, 123

•             Section 5.2 - title

•             Section 3 and line 166 are in bold

Author Response

Thank you very much for your valuable review comments. We have revised them according to your comments, please see the attachment.

Reviewer 4 Report

The paper deals with a highly relevant topic, in line with the objectives of the journal. Furthermore, the subject matter is approached with a very empirical and practical perspective, which can provide useful insights not only on a theoretical level, but above all, in terms of management inspiration for practitioners and managers. Although the paper is interesting, it requires some minor revisions before publication. Specifically, opportunities for improvement are identified in the literature review, results analysis, and conclusion sections.

In the second point of the review, important contributions have been overlooked. Here are some references to consider that contribute to the problem highlighted in the paper. In particular, it would be appropriate to consider the following works that contribute to viewing Digital Transformation not only as a process influenced by technology, but as an organizational change that impacts the thinking, behavior, and mindset of organizations:

- Schiuma, G., Schettini, E., & Santarsiero, F. (2021). How wise companies drive digital transformation. Journal of Open Innovation: Technology, Market, and Complexity, 7(2), 122.

- Santarsiero, F., Lerro, A., Carlucci, D., & Schiuma, G. (2022). Modelling and managing innovation lab as catalyst of digital transformation: theoretical and empirical evidence. Measuring Business Excellence, 26(1), 81-92.

- Tabrizi, B., Lam, E., Girard, K., & Irvin, V. (2019). Digital transformation is not about technology. Harvard business review, 13(March), 1-6.

- Santarsiero, F., Schiuma, G., Carlucci, D., & Helander, N. (2022). Digital transformation in healthcare organisations: The role of innovation labs. Technovation, 102640.

The conclusion and enlightenment should explicitly highlight the theoretical and practical contributions, research limitations, and possible future developments. Moreover, it would be interesting to understand better how the paper can provide guidance for enterprises to implement the strategy of Digital Transformation, based on the research problems illustrated in the introductory section.

Author Response

Thank you very much for your valuable review comments. We have revised them according to your comments.Please see the attachment.

Round 2

Reviewer 2 Report

Thanks for the authors' revision. Not this paper can be accepted.